# Direct and Indirect Competitive Interactions between *Ooencyrtus nezarae* and *Paratelenomus saccharalis* Parasitizing *Megacopta cribraria* Egg Patches

**DOI:** 10.3390/insects14010035

**Published:** 2022-12-30

**Authors:** Sanower Warsi, Ana M. Chicas-Mosier, Rammohan R. Balusu, Alana L. Jacobson, Henry Y. Fadamiro

**Affiliations:** 1Department of Entomology and Plant Pathology, Auburn University, Auburn, AL 36849, USA; 2Center for Environmentally Beneficial Catalysis, University of Kansas, Lawrence, KS 66045, USA; 3BASF Agricultural Products Group, Research Triangle, Durham, NC 27709, USA; 4Department of Entomology, Texas A&M University, College Station, TX 77843, USA

**Keywords:** aggressive behavior, interspecific interaction, intraguild predation, simultaneous arrival, sequential arrival

## Abstract

**Simple Summary:**

This article studied the interaction between two egg parasitoids, *Paratelenomus saccharalis* and *Ooencyrtus nezarae,* of the kudzu bug (*Megacopta cribraria*). In the laboratory, we focused on how the timing of arrival to host eggs influences egg parasitism, competition, aggressive behaviors of wasps, and intraguild predation of wasp larvae in host eggs. Results showed that interspecific interactions reduced *P. saccharalis* emergence in the presence of *O. nezarae*; however, the degree of this effect altered when wasps were released sequentially versus simultaneously and over time. Adults *P. saccharalis* competed aggressively for the shared host, although *O. nezarae* larvae outcompeted in multiparasitized eggs. Our results suggest that *O. nezarae* has the potential to negatively impact the population dynamics of *P. saccharalis*, which could influence the success of biological control programs targeting the kudzu bug.

**Abstract:**

The present study investigated egg parasitoid interspecific interactions between a generalist, *Ooencyrtus nezarae* Ishii (Hymenoptera: Encyrtidae) and a specialist, *Paratelenomus saccharalis* Dodd (Hymenoptera: Platygastridae) in a laboratory setting using kudzu bug (*Megacopta cribraria* Fabricius, (Hemiptera: Plataspidae)) eggs as their shared host. Three experiments were conducted to evaluate the emergence of wasps from parasitized hosts after the simultaneous and sequential release of wasps, monitor aggressive behavior of *P. saccharalis*, and quantify intraguild predation of *O. nezarae* larvae on heterospecific *P. saccharalis* larvae. Results showed that total host egg parasitism was higher when both wasps were released simultaneously than if wasps were released sequentially. *Ooencyrtus nezarae* produced more total offspring than *P. saccharalis* in all sequential/simultaneous treatments but produced male offspring in most cases. In the aggressive behavioral experiment, specialist, *P. saccharalis* used head butting to fight *O. nezarae,* but no other aggressions were observed. In an experiment examining intraguild predation, *O. nezarae* was able to develop in host eggs parasitized by *P. saccharalis* four days earlier, acting as a superior larval competitor. These findings shed light on the potential interspecific interactions between *O. nezarae* and *P. saccharalis,* which may determine their relative abundance and influence their compatibility in kudzu bug biological control programs.

## 1. Introduction

Adult female parasitoid wasps deposit their eggs in or on the body surface of other arthropods, which act as the host for developing larvae [1]. In general, parasitoids oviposit on an immature stage of the host, e.g., egg(s), larva(e), or pupa(e). The host is then consumed by the developing parasitoid larva(e) [1]. Parasitoid species may rely on shared common resources (e.g., host species); thus, interactions among these parasitoid species can occur frequently [2,3,4,5,6]. Researchers have debated whether the occurrence of multiple parasitoids has a beneficial or antagonistic effect on pest suppression in agricultural systems. According to some studies, multiple parasitoids might work in tandem to greatly reduce pest populations through direct competition [7,8]. Other research has shown that competition among parasitoids may impair pest suppression and disrupt biological control through indirect competition or intraguild predation of larvae [9,10]. Understanding the interspecific competition between multiple parasitoids in relation to their shared host(s) is critical to determining the efficacy of biological control programs [11,12].

Parasitoids do not immediately remove prey from the habitat like predators, and intact hosts can still be exploited by conspecific or interspecific competitors which may result in indirect competition [13]. The outcome of parasitoid competition can be characterized as direct or indirect [14]. Direct competition involves interaction between two or more individuals that utilize the same resource. Direct competition can occur when adult parasitoids search and compete for the same host resource or among immatures that develop on/in the same host (i.e. superparasitism or multiparasitism) [1,13,14,15,16]. An adult female’s direct competition with a conspecific/heterospecific female can be lethal if she engages in fighting behaviors for egg mass (patch) ownership [17]. The intensity of agonistic behaviors has been shown to escalate in specialist parasitoids because resources are limited as compared to generalists [18]. For immature parasitoids, the presence of more than one larva in a host may alter host quality, quantity, and modify the successful emergence rate of the wasp by leading to the death of the weaker competitor, or increase variability in body size among competitors [13]. As a result, the deposition of eggs in or on a parasitized host generally reduces the fecundity of ensuing adults. Dorn and Beckage [19] reported that the number of emerging adults decreases as the number of immature individuals in the host increases. This phenomenon occurs commonly in solitary parasitoids where only one larva can develop inside the host [20]. Despite this, it has been observed that parasitoids with a broad host range (i.e., generalist species) are typically more likely to exploit already parasitized hosts [21,22]. Host quality can also impact sex ratios. According to local mate competition (LMC) theory, when high-quality hosts are limited, female wasps prefer to oviposit male eggs on poor-quality hosts. In the haplodiploid mating system of parasitoids, female parasitoids produce more males if poor-quality hosts predominate, so multiparasitism and the prospects of poor host quality may result in male-biased sex ratios.

Adult parasitoid competitive success depends on patch allocation time and the decision to invest in the defense of an exploited patch or seek out an unparasitized patch [17]. Many studies have shown that the first parasitoid that oviposits generally outcompetes later individuals of competitor species [16,23]. There are two possible explanations in this case: the established larvae would have (i) consumed most of the nutrition of the host, or (ii) killed eggs or larvae and monopolized the host [1]. Parasitoid larvae may eliminate competitors through physical attack or physiological suppression [16,24]. The first instar larvae of many parasitoids possess large mandibles which may provide them with a competitive advantage over first instars that possess small mandibles [25,26]. Physiological suppression includes toxin secretion, asphyxiation, deprivation of nutrients, and hormonal interference [27]. The order of oviposition and time lag between oviposition events can also influence the competition’s outcome [16,23].

Kudzu bugs, *Megacopta cribraria* Fabricius (Hemiptera: Plataspidae), and their egg parasitoids can be used as a model host-parasitoid system to evaluate the outcome of interspecific competition. Two parasitoids, a generalist *Ooencyrtus nezarae* Ishii (Hymenoptera: Encyrtidae), and a specialist *Paratelenomus saccharalis* Dodd (Hymenoptera: Platygastridae), co-occur and parasitize kudzu bug eggs in Southeast Asia and the southeastern United States [28,29,30,31]. Immatures of both parasitoids feed and develop inside of the host eggs [32,33]. *Paratelenomus saccharalis* is a primary and solitary endoparasitoid of the Plastaspid family [34]. To date, host data of *P. saccharalis* in the United States has only been reported from *M. cribraria* eggs. Another potential host of *P. saccharalis* is *Brachyplatys subaeneuus* Westwood (Hemiptera: Plataspidae), which was reported in Miami, Florida in 2020 [35]. *Ooencyrtus nezarae* is a generalist egg parasitoid reported to attack *M. cribraria,* and pentatomids that occur in Asia, Africa, Australia, and Europe [28,36,37], and in 2016 was first reported in Alabama [38]. *Ooencyrtus nezarae* is a gregarious parasitoid in eggs of larger hosts, but in eggs of *Megacopta* spp., it is usually a solitary parasitoid [39]. In their native range of southeast Asia, *P. saccharalis* and *O. nezarae* have been reported to parasitize *M. cribraria* eggs at rates of 100% and 76.9%, respectively [28,29]. In Alabama, *P. saccharalis* and *O. nezarae* have been observed parasitizing kudzu bug eggs in the same soybean field with rates ranging from 42–95% and 82.8–100%, respectively [38], but competition between *O. nezarae* and *P. saccharalis* has not been studied in this region. During a three-year field study (2018–2020) in central Alabama, the number of *P. saccharalis* fell sharply. The population of *P. saccharalis* reached its lowest level near zero throughout the collection period since population monitoring began in 2013 [40]. A decline of *P. saccharalis* (specialist species) is concerning because this may disrupt the biocontrol of *M. cribraria*. One potential explanation is the arrival of *O. nezarae* in 2016 in Alabama [38]. Research is needed to understand how competitive interaction between these two parasitoid species impacts host suppression, and whether competitive interference may disrupt biocontrol.

The present study investigates competitive interaction through a series of experiments. Experiments were conducted to evaluate (i) the outcome of multiparasitism when *O. nezarae* and *P. saccharalis* are released simultaneously or sequentially (i.e., emerged wasp vs. host nymph), (ii) larval competition between *O. nezarae* and *P. saccharalis*, (iii) behavior of *P. saccharalis* adults when they directly interact with adult *O. nezarae* on host patches, and (iv) the outcome and consequences of intraguild predation of larvae. It is hypothesized that multiparasitism will increase larval mortality because the quality of host tissues is reduced in parasitized hosts, and will result in a decrease in fitness of surviving larvae [41]. We also hypothesized that the specialist *P. saccharalis* adults will win direct competitions against *O. nezarae* as either the prior owner of the host or the intruder because specialists are more efficient in resource utilization [42] and more likely to be aggressive over limited and valuable resources [43,44,45]. *Paratelenomus saccharalis* was also expected to outcompete *O. nezarae* during larval competition due to first instar *O. nezarae* larvae possessing small mandibles and remaining attached to their respiratory stalks within a host egg, which limits their mobility and ability to defend themselves against Platygastrid larvae that usually have large and sickle-shaped mandibles [8,46,47].

## 2. Materials and Methods

### 2.1. Plants

Soybean seeds (var. Pioneer P49T97R-SA2P) were planted into pots (15.24 cm diameter and 14.22 cm depth) in Sunshine potting mixture #8 (SunGro Horticulture, Bellevue, WA, USA) and grown in an incubator free of pests and pesticides at (26 ± 2 °C and 55 ± 5% RH) [30]. Plants were watered daily (~200 mL per pot) and fertilized according to the manufacturer’s instructions (Scotts-Sierra Horticultural Product Company, Marysville, OH, USA) once a week until use for *M. cribraria* rearing.

### 2.2. Insect Rearing

A colony of adult kudzu bugs was established by collecting insects from kudzu, *Pueraria montana* (Lour) Merr., in Auburn, AL (32.5934° N, 85.4952° W) from late March to May 2021. They were reared in ventilated plastic cages (30 cm × 30 cm × 30 cm) (BugDorm-2, Megaview Science Education Services Co., Ltd., Taichung, Taiwan) at 25 ± 1°C, 14:10 h (L:D), and 75 ± 5% RH [30] in a growth chamber (Percival, Perry, IA, USA.) and provided organic green beans and a vegetative-stage (V2-V3) soybean plant, approximately 18 to 27 cm tall, that could easily fit into the cage. New soybean plants were provided every week [30]. Cages were checked every day for fresh eggs (≤24 h) that appeared milky white in appearance as compared to aged eggs (>24 h) that were darker in color. Each experiment used ≤24 h old kudzu bug eggs (24–30 eggs).

Parasitoid species used in these experiments originated from the same kudzu patches in Auburn, AL, as described above. Collected egg masses with parasitized kudzu bug eggs (grey color) [48] were kept in a 59.1 mL portion cup (Dart container corporation, Mason, MI, USA) at 25 ± 1 °C, 14:10 (L:D) h and 75 ± 5% RH [30] until emergence. Parasitoids were identified as *O. nezarae* or *P. saccharalis*, based on details given in [38,49]. Both species were separated into different rearing cups (59.1 mL), and colonies were reared by providing adults with a honey solution (70% *v*/*v*) and allowing them to oviposit into fresh kudzu bug eggs (≤24 h old). The honey solution was held in a 0.5 mL Eppendorf microcentrifuge tube with a hole in the bottom through which a cotton string was threaded. Another microcentrifuge tube filled with water was placed to control relative humidity in each cup. This tube was perforated above the water line to dissipate moisture vapor throughout the tube to prevent insect desiccation. Approximately 20 holes were made on the cup wall with a pin for aeration and to prevent condensation. Parasitoid rearing cups were maintained at 25 ± 1°C, 14L:10D h, and 75 ± 5% RH in an incubator (Percival, Perry, IA, USA). These same conditions were used for experiments described below.

### 2.3. Physiological Condition of Insects

In all experiments, one- and four-day-old *P. saccharalis* and *O. nezarae* adult, respectively, were used [50,51]. This differential age range was chosen to coincide with each wasp’s reproductive peaks so that optimum reproduction is represented in these experiments. Within one day of emergence (egression of wasps from host eggs), individual *O. nezarae* wasp females and males were held together for 96 h to ensure mating occurred quickly after emergence [52]. Only female *P. saccharalis* have emerged from field-collected *M. cribraria* eggs. We also observed that these emerged *P. saccharalis* females were able to make offspring without mating. Therefore, in all experiments, we used unmated *P. saccharalis* females. Both species were fed 70% honey solution and were naïve (never had oviposition experience) before use in the experiments.

### 2.4. Experiment I: Timing of Adult Arrival at the Competition

This objective investigated the role of species oviposition order, and the time interval between oviposition of heterospecific females, on the outcome of parasitoid emergence. The following three combinations of wasp species introductions were tested at four-time intervals (12, 24, 48, or 72 h) each: (1) Simultaneous release: a *P. saccharalis* female and an *O. nezarae* female were released together for the total time interval (*P. saccharalis* + *O. nezarae*); (2) Sequential release i: one female *P. saccharalis* was released first for half of the total time trial, then it was removed and an *O. nezarae* female was introduced for the remaining half of the time (*P. saccharalis → O. nezarae)*; and (3) Sequential release ii: one female *O. nezarae* was released first for half of the total time trial, then it was removed, and a *P. saccharalis* female was introduced for the remaining half of the time (*O. nezarae → P. saccharalis*). Wasps from all treatments were removed from the experimental arenas (59.1 mL cup with dimensions of top diameter: 6 cm, bottom diameter: 4.4 cm, and height: 2.8 cm) after the time interval was completed. Host eggs exposed to wasps were incubated until offspring emerged as adults. The total number of live offspring was counted for each species, and unhatched or unparasitized eggs of *M. cribraria* were counted as ‘unascribed’.

For each treatment (order of oviposition × time interval), 20 replications were performed. The minimum exposure (12 h) and host egg density (24–30 kudzu bug eggs per replicate) were chosen based on the previous studies for *P. saccharalis* [51] and preliminary studies for *O. nezarae* (unpublished data) that demonstrated both species alone can parasitize at least one egg/h. Increasing host-limitation over extended periods (e.g., 72 h and 24 eggs) would increase the likelihood of interspecific competition.

### 2.5. Experiment II: Characterizing Aggressive Behavior of Parasitoids

The purpose of this experiment was to record any aggressive behavior of *P. saccharalis* or *O. nezarae* when encountering a heterospecific on the same host patch of kudzu bug eggs. One female *P. saccharalis* and *O. nezarae* were simultaneously released into a Petri dish (60 mm × 15 mm) with kudzu bug eggs. A total of 20 replications were conducted for this experiment. Their behavior was recorded for 1 h after their release in the experimental arena. This is the time (1 h) necessary for oviposition. However, these behaviors typically begin within the first 4 min (personal observation), and a female was discarded from the trial if she did not show oviposition behavior within 4 min. A handheld video camera mounted on a tripod stand was used to record the parasitoid behavior through a microscope eyepiece (Model no. SZ2-ILST, Olympus, Tokyo, Japan, magnification 5.6×). The number of confrontations were recorded and included wing waving, chasing, head butting, and boxing with forelegs [53].

### 2.6. Experiment III: Intraguild Predation of Parasitoid Larvae

Intraguild predation of *O. nezarae* on *P. saccharalis* was assessed. It was a unidirectional experiment in which *O. nezarae* was released after *P. saccharalis* because the former is a generalist species, and it has been observed that species with a broad host range generally act as hyperparasitoids that feed and oviposit on hosts and primary parasitoids [54]. A naïve female *P. saccharalis* was offered kudzu bug eggs for 24 h, and then removed. After removing *P. saccharalis*, host eggs were held for 0, 24, 48, 72, or 96 h before exposing them to a mated, naïve female *O. nezarae* for 24 h to determine larval and pupal predation on *P. saccharalis*. Eggs exposed to *P. saccharalis* were held for a maximum of 96 h because it takes 168–192 h for kudzu bug eggs to hatch, the eggs were approximately 168 h old after exposure to *O. nezarae*. After exposing *O. nezarae*, host eggs were incubated until the emergence of *P. saccharalis* and *O. nezarae* offspring. Twenty replicates were conducted for this experiment.

## 3. Data and Statistical Analysis

All statistical analyses were conducted with SAS (Ver. 9.4, SAS Institute, Cary, NC, USA). Interspecific competitive interaction results from Experiment I were analyzed using a generalized linear mixed model (GLIMMIX) with a normal distribution. The mean proportion of parasitism (parasitized eggs/total eggs), the proportion of kudzu bug nymphs (nymphs/total host eggs), the proportion of wasp offspring (total wasp offspring/total host eggs), and the sex ratio of wasps that emerged from eggs (females/total wasps) were compared among treatments. A two-way ANOVA using a simulated test for mean comparisons was used to analyze the independent variables host exposure time (12, 24, 48, and 72 h) and order of oviposition (*O. nezarae* + *P. saccharalis*, *O. nezarae* → *P. saccharalis,* and *P. saccharalis → O. nezarae*).

In Experiment II, the frequency of confrontation was calculated as the number of encounters divided by the length of the period (1 h) per replicate. All frequencies were summed together and then divided by sample size (*n* = 20) to calculate the average confrontation frequency. We also reported characterized aggressive behaviors including wing waving, chasing, head butting, and boxing with forelegs.

The same parasitism metrics were collected for Experiment III as defined for Experiment I. The independent variables of this experiment were time intervals (0, 24, 48, 72, or 96 h), wasp species (*O. nezarae* and *P. saccharalis*), and their interaction. Data were analyzed using a one way-ANOVA in Proc GLIMMIX with a normal distribution and a post hoc mean comparison was performed using the simulated test.

## 4. Results

### 4.1. Experiment I: The Timing of Adult Arrival at the Competition

Summary statistics by release order of parasitoids are provided in Table 1. Order of release influenced the host parasitism and strength of interspecific competition, represented by the wasp emergence. The proportion of parasitized eggs was the highest when *P. saccharalis* arrived earlier or together with *O. nezarae*. However, the proportion of emerged wasps from parasitized eggs was significantly higher when both competitors arrived simultaneously at the host patch, intermediate when *P. saccharalis* arrived first, and lowest when *O. nezarae* arrived before *P. saccharalis* (for statistics, see Table 1).

Out of 6147 *M. cribraria* eggs, a total of 2501 were parasitized. The proportion of parasitized host eggs was affected by the order of the two competitors’ release and the period after they first oviposited (Figure 1). Over the entire experiment, the distribution of parasitoid attack was generally higher with increasing host exposure time, with the exception of *O. nezarae* to *P. saccharalis*, indicating that prolonged exposure time provides parasitoids an advantage in parasitizing more host eggs (Figure 1). The increase in parasitism also resulted in a decrease in kudzu bug nymph survival (Figure 2). Interestingly, the host eggs where *P. saccharalis* first oviposited and were followed by *O. nezarae* showed higher mortality with longer exposure times (Figure 1).

*Ooencyrtus nezarae* showed differences in its emergence when competing against *P. saccharalis*. From a total of 1893 parasitoids, *O. nezarae* yielded an average of 0.74 offspring. The emergence of *O. nezarae* offspring was higher when their oviposition was simultaneous with *P. saccharalis* (Figure 3). It is noteworthy that *O. nezarae* emergence was relatively higher when the host eggs were exposed to both parasitoids for a prolonged period of 48–72 h (Figure 3). In addition, the proportion of *O. nezarae* offspring was around one-fold higher when *O. nezarae* arrived second to the host and when there was a delay of more than 12 h between oviposition of *P. saccharalis* and *O. nezarae*. It indicated that *O. nezarae* won almost all competitive events in the parasitized hosts.

The adult emergence for *P. saccharalis* was different from *O. nezarae*. A proportion of 0.26 *P. saccharalis* emerged from 1893 parasitized host eggs. The interaction between *O. nezarae* and *P. saccharalis* was not favorable for *P. saccharalis* offspring when *P. saccharalis* arrived at the host patch later or simultaneous with *O. nezarae.* In most cases, *P. saccharalis* emergence was highly affected by interference competition when it was introduced later, resulting *in P. saccharalis* having fewer offspring emergence. It indicates that *P. saccharalis* accepted a lower number of host eggs previously exposed to *O. nezarae* at all time points (Figure 4). Emergence of fewer *P. saccharalis* offspring suggested that *P. saccharalis* has a competitive disadvantage when it does not have an opportunity to develop to the first instar larva prior to multiparasitism by *O. nezarae*. A proportion of 0.20 kudzu bug eggs remained unhatched when *P. saccharalis* and *O. nezarae* arrived at the host patch together. However, a higher proportion of eggs remained unhatched when competitors arrived in sequence. *Ooencyrtus nezarae’s* arrival at the host patch first resulted in proportions of unhatched eggs exceeding 0.40 (Appendix A), indicating potential feeding behavior on the host.

The proportion of females in *O. nezarae* offspring emerging from parasitized eggs was correlated with the interaction of the order of parasitism and time interval (for statistics, see below Figure 5). *Ooencyrtus nezarae* females appeared to adjust the sex allocation of their progeny in response to competition. In most cases, multiparasitism corresponded to the *O. nezarae* population having a male-biased sex ratio (Figure 5). Even though *P. saccharalis* only produced female offspring from the field, the number of females in *P. saccharalis* offspring was higher when *P. saccharalis* had the opportunity to arrive at the host patch earlier than their competitors (Figure 4).

### 4.2. Experiment II: Characterizing Aggressive Behavior

*Ooencyrtus nezarae* and *P. saccharalis* showed differences in their behavior when competing against each other. When both wasp species were using the host egg patch at the same time, interspecific aggressive behavior occurred between the two species. As soon as *P. saccharalis* encountered or noticed *O. nezarae* on the egg mass, they displayed aggressive behaviors, which occurred at a frequency of 1.85 ± 0.25 h^−1^. The *P. saccharalis* employed head butting (37 times in 20 replications) to fight the *O. nezarae* in all their confrontations, and it occasionally flapped its wings (2 times in 20 replications), appearing ready to strike, then charged its competitor, causing it to leave the egg mass. No aggressive behavior was observed by *O. nezarae*. Female *O. nezarae* displayed two behaviors on approaching or encountering *P. saccharalis:* running, in which the *O. nezarae* female walked away from the *P. saccharalis*, and left the host patch, and avoiding, in which the *O. nezarae* avoided physical contact with the approaching *P. saccharalis* by changing her searching direction.

### 4.3. Experiment III: Intraguild Predation of Parasitoid Larvae

The summary statistics of intraguild interactions between *O. nezarae* and *P. saccharalis* are summarized in Table 2. Intraguild predation of *O. nezarae* larvae often did not result in a decrease in the host population. *A* higher impact on the host was mostly achieved when *O. nezarae* was released right after the *P. saccharalis* were released. However, the proportion of host parasitism dropped from 0.23 to 0.19 in eggs that were exposed to *P. saccharalis* prior to being exposed to *O. nezarae*, 72 h or more before (Table 2).

The proportion of wasp emergence was also affected by intraguild predation of *O. nezarae* larvae (Table 2). The wasp emergence decreased from the parasitized eggs as the differences in the timing of oviposition between both parasitoids increased except for the time delay of 24 h. *Ooencyrtus nezarae* accepted all host eggs that had been exposed to *P. saccharalis* from 0–96 h before. The offspring of *O. nezarae* was even able to develop in host eggs parasitized by *P. saccharalis* 96 h earlier (Figure 6).

In most of the situations, the sex ratio of *O. nezarae* was male-biased and there was a trend towards longer duration experiments showing less favorable effects on female emergence of *O. nezarae* (Figure 7).

## 5. Discussion

We investigated both the direct and indirect interspecific interference competition between *P. saccharalis* and *O. nezarae* by comparing the proportion of parasitized host eggs, host nymphs, and emerged parasitoids in a sequential or simultaneous release of both species. Both wasp species showed differences in host parasitism and emergence when competing against each other. *Ooencyrtus nezarae* acted as a superior larval competitor in almost all competitive events. *Paratelenomus saccharalis* adults showed agonistic behavior against its heterospecific female to defend the host patch. The outcomes of such competitive situations are discussed below.

### 5.1. Experiment I: The Timing of Adult Arrival at the Competition

The direct interference results showed that competition occurs between both parasitoid species for the common host, and the order of release, i.e., the timing of the wasp’s arrival at the host patch, also influenced the outcome. When specialist *P. saccharalis* arrived earlier or together with generalist *O. nezarae*, egg parasitism rates were the highest. However, when *O. nezarae* exploited the host patch first, the overall parasitism was lower. Our results are consistent with those of [55] showing that specialist (*Microplitis mediator* Haliday) had more impact on the parasitism of *Helicoverpa armigera* Hübner larvae when it was released prior to a generalist (*Campoletis chlorideae* Uchida) [55]. The possible explanation for these results is that the specialist species has better host handling strategies due to specificity, and is much more efficient in host utilization compared to a generalist [56]. The host handling time of female *P. saccharalis* is much shorter (an average of 10.48 min) [51] than *O. nezarae* (an average of 19.32 min) [39].

The seasonal arrival time of *P. saccharalis* is quite different in nature. In the native range (Japan), *P. saccharalis* arrives in May and *O. nezarae* in June [31]. In the United States, *O. nezarae* arrives earlier in May, whereas *P. saccharalis* first appears in July, and both species overlap from July to October in soybean fields (personal observation). *Ooencyrtus nezarae* host feeding behavior may also influence parasitism. When *O. nezarae* was released first, regardless of host exposure time, the highest proportion of kudzu bug eggs were neither hatched nor parasitized (i.e., unascribed eggs) by wasps. The proportion of unascribed eggs was equal to or more than that of parasitized eggs.

Parasitism and emergence were highest when *O. nezarae* and *P. saccharalis* arrived at the host patch simultaneously, and only a small fraction of parasitized eggs did not produce wasps in this treatment. When *O. nezarae* arrived later than *P. saccharalis*, the proportion of parasitized host eggs was higher, but fewer wasps emerged from parasitized eggs. More than half of the parasitized eggs did not yield parasitoids and were non-viable (Appendix A). A similar result was found when *O. nezarae* arrived earlier than *P. saccharalis* at the host patch; the highest proportion of parasitized eggs that did not produce wasps was observed when the wasp was given a longer time to exploit the host (Appendix A). These results might be related to the preference of *O. nezarae* for parasitized eggs over unparasitized eggs. *Ooencyrtus nezarae* female prefers parasitized host eggs to save their energy and time in host drilling [39]. In fact, the handling time of *O. nezarae* on a parasitized host is an average of 17.23 min, which is considerably shorter than an unparasitized host, taking only 19.32 min in handling [39]. It also suggests that *O. nezarae* is superior in interspecific larval competition (intrinsic competition).

### 5.2. Experiment II: Characterizing Aggressive Behavior

When *P. saccharalis* and *O. nezarae* arrive at the host patch together, *P. saccharalis* exhibits aggressive behavior; *O. nezarae* did not show any distinct behavior toward *P. saccharalis* to defend the kudzu bug egg patch. It was observed that *O. nezarae* reached the host eggs earlier than *P. saccharalis* and took possession of the eggs. Then, *P. saccharalis* females that arrived later would fight with *O. nezarae* to access the host. *Paratelenomus saccharalis* exhibited continuous head striking of *O. nezarae* that generally caused *O. nezarae* to leave the host patch. Such aggressive tactics were also observed in other Platygastrid egg parasitoids such as *Trissolcus basalis* Wollaston against *O. telenomicida* Vassiliev to utilize *Nezara viridula* Linnaeus eggs [47]. In Japan, *P. saccharalis* females were also observed to aggressively exclude females of *O. nezarae* from utilizing the host patch [31]. Since *O. nezarae* can utilize both unparasitized and parasitized hosts, aggressive behavior from *O. nezarae* towards *P. saccharalis* is not advantageous, therefore leaving the host patch and searching out additional patches without a *P. saccharalis* female present would be expedient for them. The lack of aggressive behavior in *O. nezarae* females may also be related to their broad host range in the United States [38], or due to its smaller size (max. overall body length = 0.77 mm) in comparison to *P. saccharalis* females (max. overall body length 0.83 mm) [30,34].

### 5.3. Experiment III: Intraguild Predation of Larvae

Intraguild predation data reaffirms the vulnerability of *P. saccharalis* immatures (72–96 h), as *O. nezarae* successfully emerges from multiparasitized eggs, speculating the adaptive outcome of the temporal trophic shift from *M. cribraria* eggs to its primary parasitoid, *P. saccharalis*. It increased the window of opportunity to exploit host eggs and allowed them to evade exclusion by the more fecund wasp species. However, this trophic shift ended with a detrimental outcome for both species, as non-reproductive mortality increased significantly (0.84 at 96 h). Our result was consistent with Cusumano and Peri [47], who observed high levels of dead host and parasitoid mortality with later-stage multiparasitism by *O. telenomicida*.

It has been observed that *Ooencyrtus* spp. with a broad host range generally shift their trophic position when hosts become scarce [21,22,57]. Mohammadpour et al. [8] investigated competition between *O. pityocampi* Mercet and *Trissolcus agriope* Kozlov and Le on the host eggs of *Brachynema signatum* Jakovlev. *Ooencyrtus pityocampi* was able to develop as a superior larval competitor or could be as a facultative hyperparasitoid on the latter species [8]. This is presumably through direct physical attack between larvae [58]. Studies also show that embryological differences between the species can be one possible reason to outcompete the competitor by hatching earlier [1]. However, the developmental biology of *P. saccharalis* and *O. nezarae* have yet to be investigated. 

## 6. Conclusions

Both parasitoid species differed in terms of their host utilization and competitive interference strength. *Paratelenomus saccharalis* was the species that had the greatest ability to exploit the resource, while *O. nezarae* was the strongest species in the direct and indirect competition. Our work has shown that *O. nezarae* has the potential to impact the population dynamics of *P. saccharalis*, which could be detrimental to biological control programs of *M. cribraria*. Additional field studies, however, are needed to determine interspecific competition under natural conditions.

## Figures and Tables

**Figure 1 insects-14-00035-f001:**
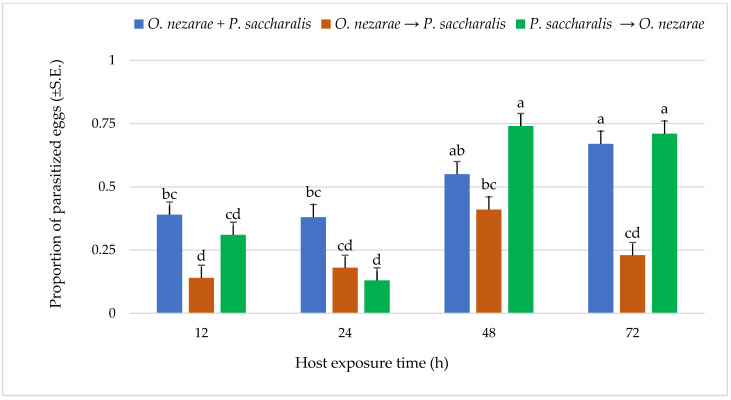
LS-mean (±S.E) proportion of parasitized *Megacopta cribraria* eggs under different parasitoid timing x host exposure time conditions. Scenarios examined included simultaneous release of *Ooencyrtus nezarae* and *Paratelenomus saccharalis* (*O. nezarae + P. saccharalis*), sequential release in which *O. nezarae* was allowed to exploit the host patch first (*O. nezarae → P. saccharalis*), or sequential release in which *P. saccharalis* was released first (*P. saccharalis → O. nezarae*). Data were graphed with two-way ANOVA using simulated multiple comparison test: Release order (F = 31.45, df = 2, 228, *p* < 0.0001), host exposure time (F = 35.06, df = 3, 228, *p* < 0.0001) and their interaction (F = 6.86, df = 6, 228, *p* < 0.0001). Different letters indicate a significant difference at *p* < 0.05 for the interaction term. (Sample size, *n* = 20).

**Figure 2 insects-14-00035-f002:**
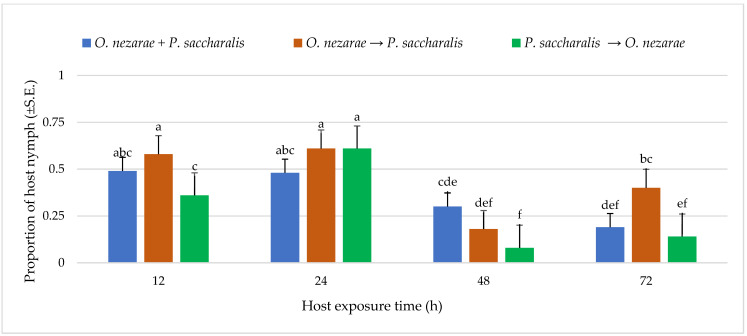
LS-mean (±S.E) proportion of *Megacopta cribraria* nymph hatching from eggs in the timing of adult arrival at the competition experiment. Scenarios examined included simultaneous release of *Ooencyrtus nezarae* and *Paratelenomus saccharalis* (*O. nezarae + P. saccharalis*), sequential release in which *O. nezarae* was allowed to exploit the host patch first (*O. nezarae → P. saccharalis*), or sequential release in which *P. saccharalis* was released first (*P. saccharalis → O. nezarae*). Data were graphed with two-way ANOVA using simulated multiple comparison test: host exposure time (F = 51.35, df = 3, 228, *p* < 0.0001), release order (F = 11.24, df = 2, 228, *p* < 0.0001), and their interaction (F = 4.96, df = 6, 228, *p* < 0.0001). Different letters indicate a significant difference at *p* < 0.05 for the interaction term. (Sample size, *n* = 20).

**Figure 3 insects-14-00035-f003:**
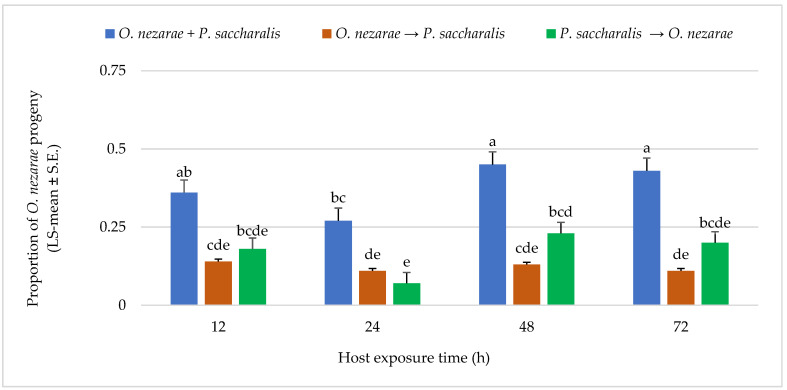
LS-mean (±S.E) of *Ooencyrtus nezarae* progeny that emerged from *Megacopta cribraria* eggs (Sample size, *n* = 20) in the timing of adult arrival at the competition experiment. Scenarios examined included simultaneous release of *O. nezarae* and *Paratelenomus saccharalis* (*O. nezarae + P. saccharalis*), sequential release in which *O. nezarae* was allowed to exploit the host patch first (*O. nezarae → P. saccharalis*), or sequential release in which *P. saccharalis* was released first (*P. saccharalis → O. nezarae*). Data were graphed with two-way ANOVA using simulated multiple comparison test: Release order (F = 63.78, df = 2, 228, *p* < 0.0001), host exposure time (F = 7.51, df = 3, 228, *p* < 0.0001) and their interaction (F = 1.77, df = 6, 228, *p* < 0.0001). Different letters indicate a significant difference at *p* < 0.05 for the interaction term.

**Figure 4 insects-14-00035-f004:**
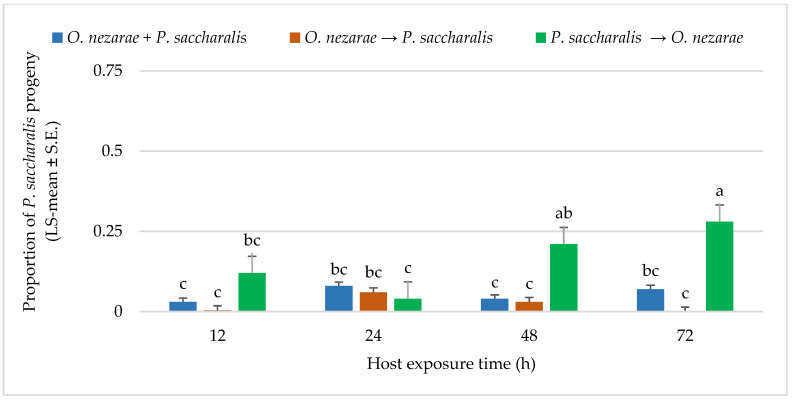
LS-mean (±SE) of *Paratelenomus saccharalis* progeny that emerged from *Megacopta cribraria* eggs (Sample size, *n* = 20) in the timing of adult arrival at the competition experiment. Scenarios examined included simultaneous release of *Ooencyrtus nezarae* and *P. saccharalis* (*O. nezarae + P. saccharalis*), sequential release in which *O. nezarae* was allowed to exploit the host patch first (*O. nezarae → P. saccharalis*), or sequential release in which *P. saccharalis* was released first (*P. saccharalis → O. nezarae*). Data were graphed with two-way ANOVA using simulated multiple comparison test: Release order (F = 20.23, df = 2, 228, *p* < 0.0001), host exposure time (F = 2.68, df = 3, 228, *p* = 0.047) and their interaction (F = 1.77, df = 6, 228, *p* = 0.0002). Different letters indicate a significant difference at *p* < 0.05. for the interaction term.

**Figure 5 insects-14-00035-f005:**
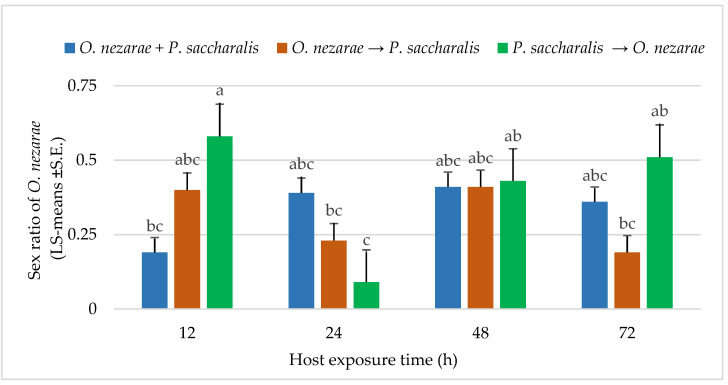
LS-mean (±S.E.) sex ratio (females/total progeny) of *Ooencyrtus nezarae* (Sample size, *n* = 20) on *Megacopta cribraria* eggs in the timing of adult arrival at the competition experiment. Scenarios examined included simultaneous release of *O. nezarae* and *Paratelenomus saccharalis* (*O. nezarae + P. saccharalis*), sequential release in which *O. nezarae* was allowed to exploit the host patch first (*O. nezarae → P. saccharalis*), or sequential release in which *P. saccharalis* was released first (*P. saccharalis → O. nezarae*). Data were graphed with two-way ANOVA using simulated multiple comparison test: Release order (F = 1.82, df = 2, 228, *p* = 0.16), host exposure time (F = 3.61, df = 3, 228, *p* = 0.01) and their interaction (F = 4.86, df = 6, 228, *p* = 0.0001). Different letters indicate a significant difference at *p* < 0.05 for the interaction term.

**Figure 6 insects-14-00035-f006:**
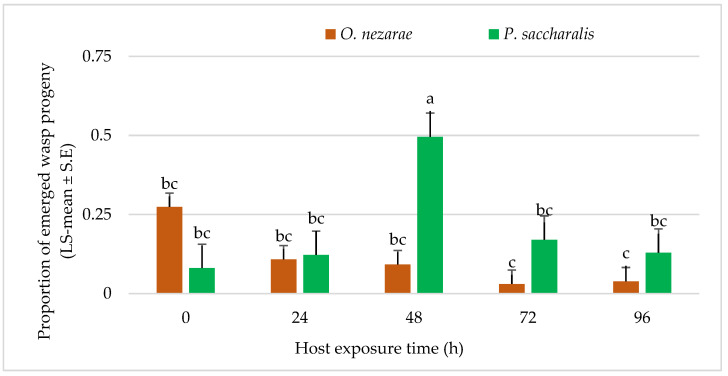
LS-mean (±SE) of *Ooencyrtus nezarae* and *Paratelenomus saccharalis* progeny that emerged from host eggs among treatments. Data were graphed with one-way ANOVA using simulated multiple comparison test (F =10.91, df =4, 95, *p* < 0.0001 for the main effect i.e., host exposure time). Different letters indicate a significant difference at *p* < 0.05. (Sample size, *n* = 20).

**Figure 7 insects-14-00035-f007:**
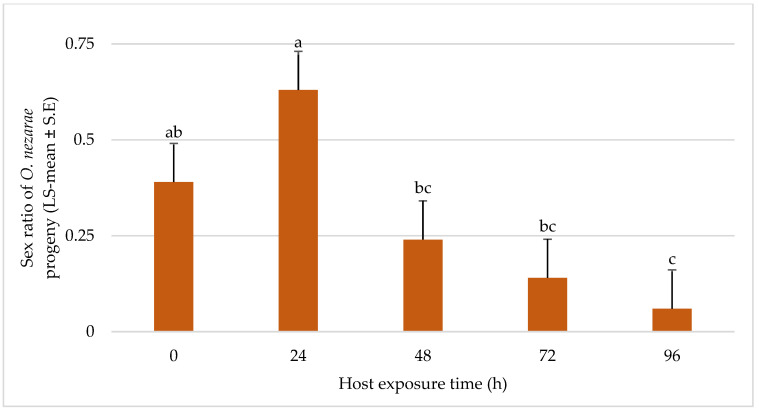
LS-mean (±S.E.) sex ratio (females/total progeny) of *Ooencyrtus nezarae* (Sample size, *n* = 20) on *Megacopta cribraria* eggs intraguild predation experiment. Data were graphed with one-way ANOVA using simulated multiple comparison test (F = 10.91, df = 4, 95, *p* < 0.0001 for the main effect i.e., host exposure time). Different letters indicate a significant difference at *p* < 0.05.

**Table 1 insects-14-00035-t001:** The proportion of parasitized host eggs and emerged wasps averaged across treatments in experiments conducted to examine how the order of *Ooencyrtus nezarae* and/or *Paratelenomus saccharalis* adult arrival at a *Megacopta cribraria* egg patch influences competition.

(Proportion ± SE)		
Treatment	*M. cribraria* Parasitized Eggs	Emerged Wasp
Simultaneous release	0.50 ± 0.025 ^a^	0.43 ± 0.02 ^a^
Sequential release with *O. nezarae* first	0.24 ± 0.025 ^b^	0.15 ± 0.02 ^c^
Sequential release with *P. saccharalis* first	0.47 ± 0.025 ^a^	0.34 ± 0.02 ^b^
ANOVA for the main effect	F = 19.91; df = 2, 237; *p* < 0.0001	F = 32.32; df = 2, 237; *p* < 0.0001

LS-means within a column followed by the same letter are not significantly different at *p* < 0.05. Data were presented with one-way ANOVA using simulated multiple comparison test.

**Table 2 insects-14-00035-t002:** The proportion of parasitized host eggs and emerged wasps averaged across treatments in an experiment examining intraguild predation of larvae experiment. Host exposure time (h) indicates the interval between the removal of a *Paratelenomus saccharalis* female and the introduction of a *Ooencyrtus nezarae* female.

(Proportion ± S.E)		
Host Exposure Time (h)	*M. cribraria* Parasitized Eggs	Emerged Wasp
**0**	0.38 ± 0.07 ^ab^	0.35 ± 0.06 ^ab^
**24**	0.35 ± 0.07 ^ab^	0.23 ± 0.06 ^b^
**48**	0.59 ± 0.07 ^a^	0.58 ± 0.06 ^a^
**72**	0.23 ± 0.07 ^b^	0.20 ± 0.06 ^b^
**96**	0.19 ± 0.07 ^b^	0.16 ± 0.06 ^b^
**ANOVA for main effect**	F = 5.21; df = 4, 95; *p* = 0.0008	F = 7.90; df = 4, 95; *p* < 0.0001

LS-means within a column followed by the same letter are not significantly different at *p* < 0.05. Data were presented with one-way ANOVA using simulated multiple comparison test.

## Data Availability

Data will be provided upon request.

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
