# Peer review of "Direct and Indirect Competitive Interactions between Ooencyrtus nezarae and Paratelenomus saccharalis Parasitizing Megacopta cribraria Egg Patches"

_insects, 2022, doi:10.3390/insects14010035_

Round 1

Reviewer 1 Report

The study titled “Direct and indirect competitive interactions between Ooencyrtus nezarae and Paratelenomus saccharalis parasitizing Megacopta cribraia eggs patches” submitted by Warsi et al. is original and novelty. It was performed under laboratory conditions with 3 main points; a) evaluating emergence of two egg parasitoid species released simultaneous and sequentially, b) evaluating aggressive response of each parasitoid, and c) determine the intraguild predation between larvae.

General comments: The present study is original. Most studies using egg parasitoids against Hemipteran (including Homoptera) generally investigate the egg parasitoid response using just one parasitoids species as model under laboratory conditions. In natural conditions several studies have found that eggs are attacked by two or more egg parasitoids. However, little is known about how this occurs under lab controlled conditions. The present study focus in two egg parasitoids within 3 main points mentioned already.

I have 4 general observations.

A) The replicate size in each of the 3 experiments is 20, however this sample size maybe is low and I am not sure about the normal distribution of the data, although they used GLIMMIX, please could you explain why the sample size is the appropriated?.

B) The proportion of the parasitized eggs, which was showed in Fig 1, how was evaluated? Observing the grey color in the parasitized eggs? Please explain in materials and methods how was determined the parasitized eggs. The parasitized eggs in Hemiptera change of color throughout time, maybe using only emergence rate of wasps is better than using proportion of parasitized eggs.

C) In the experiment II (aggressive response), results only describe the behavior, if possible will be great show that information using figures or tables in order to see the trends and the variation in this experiment II. In the manuscript say “The number of confrontations were recorded and included wing waving, chasing, head butting, and boxing with forelegs [53]. A total of 20 replications were conducted for this experiment” (lines 203-204).

D) In results section some sentences suggest speculation for instance (lines 245-247) say:  “Parasitoids did not emerge from all parasitized eggs suggesting that either due to intrinsic competition or parasitization by either parasitoid, wasps could not complete development inside the host egg”. Maybe is appropriate remove from the result section or move to discussion. The same occurs in lines 251-252. And line 373. I suggest remove any speculative sentence within this results section.     

Specific comments

A)    All the figure 6 is not observed in the submitted PDF.

B)    In the introduction section will be great if you mention that larva of each egg parasitoid develops outside the eggs? This comment is because in several egg parasitoids the larva develops within the egg. That basic information is related with the experiment III.

C)In the experiment I, the experimental arena was a petri dish (60 mm x 15 mm)??, please include that information.

Author Response

Please see the attached response.

Reviewer 2 Report

Dear Dr. Jacobson,

I have carefully read your manuscript entitled "Direct and indirect competitive interactions between Ooencyrtus nezarae and Paratelenomus saccharalis parasitizing Megacopta cribraria egg patches". I believe this manuscript contains new important information on the ecological interactions of the two parasitoid species attacking the same host, and therefore your paper could be published in Insects. I have just a few suggestions regarding the overall quality of the manuscript. Specifically, the genus name of a particular parasitoid is "Paratelenomus", not "Paratelonomus" (lines 88 and 323). Moreover, only a fragment of your Fig. 6 can be seen in the PDF version of the paper. In addition, references no. 46 to 48 are missing from the list. Finally, please remove redundant indications of DOIs from the reference list, and, at the same time, add "https://doi.org/" at the lines 594 and 649. Please also make the designations of time intervals in Table S2 uniform, e.g., "12-hour", "24-hour", etc.

Author Response

Please see the attached response.

Round 2

Reviewer 1 Report

The manuscript was sufficiently improved.